# Canonic Signed Spike Coding for Efficient Spiking Neural Networks

## Abstract

Spiking Neural Networks (SNNs) aim to mimic the spiking behavior of biological neurons and are expected to play a key role in neural computing and artificial intelligence. Converting Artificial Neural Networks (ANNs) to SNNs is a widely used approach to achieve comparable performance on large-scale datasets, with efficiency determined by acitivation encoding. Current schemes, which typically rely on spike count or timing, exhibit a linear relationship between encoding precision and the number of required timesteps. To enhance encoding capacity with reduced timesteps, we propose the Canonic Signed Spike (CSS) coding scheme. Spikes are assigned different weights during the neuron's decoding stage, maintaining a single-bit spike representation. We analyze the residual errors during encoding and introduce the Over-Fire-and-Correct (OFC) method to enable efficient computation with weighted spikes. The optimal threshold derived from our method can also be applied to integrate-and-fire (IF) neurons and improve accuracy in rate coding. We evaluate the proposed methods on the CIFAR-10 and ImageNet datasets. The experimental results demonstrate that the CSS coding scheme significantly compresses timesteps with minimal conversion loss and offers an energy efficiency advantage for the resulting SNNs.

## 1. Introduction

Spiking Neural Networks (SNNs), recognized as the third generation of neural network models, are inspired by the biological structure and functionality of the brain (Maass, 1997). Unlike traditional Artificial Neural Networks (ANNs), which rely on continuous activation func-

tions, SNNs utilize discrete spiking events. This enables SNNs to capture temporal dynamics and process information in a manner that closely resembles brain activity (Taherkhani et al., 2020). The event-driven nature of SNNs aligns with the brain's energy-efficient computational paradigm, offering potential for more efficient and low-power computing systems (Yamazaki et al., 2022).

The two primary learning algorithms for SNNs are gradient-based optimization and ANN-SNN conversion. Directly training using supervised backpropagation is challenging due to the non-differentiable nature of spike generation (Neftci et al., 2019; Lee et al., 2016; Wu et al., 2018; Bellec et al., 2019). The conversion-based method, however, offers a practical approach to overcome this difficulty and has produced the best-performing SNNs (Ding et al., 2021; Bu et al., 2022; Deng & Gu, 2021).

The core principle of ANN-SNN conversion is the encoding of ANN activations into spike train representations. Specifically, by maintaining identical weight parameters, spiking neuron models are designed to generate spike patterns that correspond directly to the ANN activations. Various coding schemes, such as rate coding and temporal coding, have been proposed to describe neural activity (Johansson & Birznieks, 2004; Thorpe & Gautrais, 1998; Gollisch & Meister, 2008). Rate coding maps the number of spikes to the activation values (Rueckauer et al., 2017; Cao et al., 2015). In contrast, temporal coding focuses on the precise timing or patterns of spikes (Yang et al., 2023; Han & Roy, 2020). For example, Time-to-First-Spike (TTFS) coding maps the the ANN activation to the time elapsed before the first spike (Stanojevic et al., 2022).

However, using spike counts or temporal duration for encoding establishes a linear relationship between encoding precision and the number of timesteps. This inherently limits the performance of converted SNNs under low timestep conditions. Recent works have proposed alleviating this problem by quantizing the ANN before conversion (Hu et al., 2023; Bu et al., 2023; Hao et al., 2023). This simplifies the activation encoding but introduces additional quantizing and training overhead. Our goal is to develop a novel encoding paradigm that enables direct conversion of full-precision ANNs while maintaining high performance at low timesteps. Notably, the proposed encoding scheme can also convert

---

[1]Anonymous Institution, Anonymous City, Anonymous Region, Anonymous Country. Correspondence to: Anonymous Author <anon.email@domain.com>.

Preliminary work. Under review by the International Conference on Machine Learning (ICML). Do not distribute.

Figure 1. Comparison of different neural coding schemes. $\mathcal{C}$ denotes the information encoded in a spike, $u$ represents the membrane potential, and $v_{th}$ is the threshold. (a) Rate coding: The encoding capacity is limited, requiring more timesteps to improve precision. (b) Directly transmitting weighted spikes: This approach requires a larger bit width to represent weight information, with neurons adjusting thresholds based on the weights, which increases network complexity. (c) Weighting stepwise during the decoding process: Our method enhances the encoding capacity of the spike train while preserving network simplicity.

quantized ANNs and further reduce the required number of timesteps.

Information theory provides a principled approach to quantify differences between encoding schemes in neural coding analysis. For instance, the encoding capacity can be measured by the number of bits that can be encoded within given timesteps (Borst & Theunissen, 1999; Panzeri et al., 2007). To enhance the expressiveness of spike trains, we introduce a temporally structured weight pattern with exponential decay. For a spike train of $T$ timesteps, rate coding or TTFS coding achieves $\lfloor \log_2(T + 1) \rfloor$ bits of encoding capacity. In contrast, our method enhances this to $\left\lfloor \log_2 \left( \sum_{t=1}^{T} \omega_t + 1 \right) \right\rfloor$ bits through the application of weights $\omega_t$, as illustrated in Figure 1 (a) and (c). We refer to these spikes as canonical due to the fixed weight pattern.

Weighting is implemented progressively during the neuron's decoding process. Specifically, at each timestep, neurons amplify the residual membrane potential by a fixed coefficient before integrating new inputs. Compared to directly integrating weighted inputs (Stöckl & Maass, 2021; Rueckauer & Liu, 2021), this approach reduces information flow and maintains a constant firing threshold, as shown in Figure 1 (b) and (c).

While the decaying weight pattern enables fast information transmission, it results in residual information accumulation in the membrane potential, which we refer to as residual errors in encoding. To address this, we propose an Over-Fire-and-Correct (OFC) mechanism: the neuron's firing threshold is reduced, while negative spikes are introduced to compensate for information overflow. Moreover, the optimal threshold we proposed can also be applied to Integrate-and-Fire (IF) neurons, reducing encoding loss in rate coding scenarios.

Based on these characteristics, we term the proposed method Canonic Signed Spike (CSS) coding scheme and the corresponding neuron model Ternary Self-Amplifying (TSA) neuron. The main contributions of this paper can be summarized as follows:

- We compress the timesteps to logarithmic scale by weighting the spikes. We propose that the neurons stepwise amplify the membrane potential by a fixed coefficient to perform the weighting. This results in a more hardware-friendly network architecture.

- We systematically analyze the residual errors during conversion and propose the OFC method for efficient neural computation. Compared to previous implementations of weighted spikes, we reduce the network output latency by a factor of $T$, where $T$ denotes the requried timesteps per layer.

- We demonstrate the effectiveness of the CSS coding scheme on the CIFAR-10 and ImageNet datasets. The results show that the proposed method reduces both the number of timesteps and conversion loss. Additionally, the CSS coding scheme offers energy efficiency advantages over both rate coding and temporal coding.

## 2. Preliminaries

In this section, we provide a concise overview of the principles for ANN-SNN conversion and demonstrate the impact of encoding schemes on its performance.

ANN-SNN conversion typically involves the following two key steps: 1) designing an encoding method to map ANN activations to spike trains, and 2) designing a suitable neuron model to ensure the generated spike train accurately encode the activation value. The most widely used and State-Of-The-Art (SOTA) approaches (Rueckauer et al., 2017; Hu et al., 2023; Hao et al., 2023) employ (signed) soft-reset IF neurons and interprets their output through spike rates.

### 2.1. Spiking Neurons

Spiking neurons communicate through spike trains and are interconnected via synaptic weights. Each incoming spike

*Table 1.* Common symbols in this paper.

| Symbol | Definition | Symbol | Definition | Symbol | Definition |
|--------|-----------|--------|-----------|--------|-----------|
| $l$ | layer index | $\theta^l$ | spike amplitude | $o_i^l[t]$ | membrane potential before reset |
| $i, j$ | neuron index | $S_i^l[t]$ | spike sequence | $z_i^l[t]$ | integrated inputs (PSP)[1] |
| $T$ | timesteps for encoding | $v_{th}^l$ | firing threshold | $w_{ij}^l, b_i^l$ | SNN weight and bias |
| $\beta$ | amplification coefficient | $u_i^l[t]$ | membrane potential after reset | $\hat{w}_{ij}^l, \hat{b}_i^l$ | ANN weight and bias |

[1] Postsynaptic potential

contributes to the postsynaptic neuron's membrane potential, and a spike is generated when the potential reaches a predefined threshold. Generally, a spike sequence $S_i^l[t]$ can be expressed as follows:

$$S_i^l[t] = \sum_{\tau \in \mathbb{F}_i^l} \theta^l \delta[t - \tau] \quad (1)$$

where $i$ is the neuron index, $l$ is the layer index, $\theta^l$ is the spike amplitude, $\delta[\cdot]$ denotes an unit impulse[1], $f$ is the spike index, and $\mathbb{F}_i^l$ denotes a set of spike times which satisfies the firing condition:

$$\tau : o_i^l[\tau] \geq v_{th}^l \quad (2)$$

where $o_i^l[t]$ denotes the membrane potential before reset and $v_{th}^l$ denotes the threshold. For soft-reset IF neuron model, the membrane potential is subtracted by an amount equal to the spike amplitude for reset. Its dynamics can be expressed as follows:

$$u_i^l[t] = u_i^l[t-1] + z_i^l[t] - S_i^l[t] \quad (3)$$

where $u_i^l[t]$ denotes the membrane potential after reset and is referred to as the residual membrane potential. $z_i^l[t]$ denotes the integrated inputs:

$$z_i^l[t] = \sum_j w_{ij}^l S_j^{l-1}[t] + b_i^l \quad (4)$$

where $w_{ij}^l$ is the synaptic weight and $b_i^l$ is the bias. For clarity, the definitions of common symbols are provided in Table 1.

### 2.2. Activation Encoding

Let $T$ denote the number of timesteps, with the initial condition $u_i^l[0] = 0$, we can iteratively update the membrane potential using Equation (3) until $t = T$. Then substitute $z_i^l[t]$ with Equation (4), and we can write:

$$\frac{\sum_t S_i^l[t]}{T} = \sum_j w_{ij}^l \frac{\sum_t S_j^{l-1}[t]}{T} + \sum_{t=1}^{T} \frac{b_i^l}{T} - \frac{u_i^l[T]}{T} \quad (5)$$

---
[1] $\delta[t]$ takes the value 1 at $t = 0$, and 0 otherwise

See Appendix A.1 for a detailed derivation. Note that both sides of the equation are divided by $T$ to better highlight the interpretation of $\sum_{t=1}^{T} S_i^l[t]/T$ as "rate". Equation (5) defines the relationship between neuron's input rate and output rate and can be directly related to the forward pass in a ReLU-activated ANN:

$$a_i^l = \max\left(\sum_j \hat{w}_{ij}^l a_j^{l-1} + \hat{b}_i^l, 0\right) \quad (6)$$

where $a_i^l$ denotes the ANN activation, $\hat{w}_{ij}^l$ and $\hat{b}_i^l$ denote the weight and bias, respectively. Note that in Equation (5) we have: 1) $\sum_t S_i^l[t]/T > 0$, and 2) $u_i^l[T]/T$ becomes negligible as $T$ increases. These observations suggest that mapping ANN activations to SNN spike rates can be achieved by simply setting $w_{ij}^l = \hat{w}_{ij}^l$ and $b_i^l = T\hat{b}_i^l$.

However, with fewer time steps, the spike rate $\sum S_i^l[t]/T$ can only encode a limited number of activations, leading to a rapid increase in conversion loss. This issue stems from the linear scaling of spike count $\sum_t S_i^l[t]$ with timesteps, where each additional timestep provides only a constant information gain. Therefore, our goal is to incorporate nonlinearity into the encoding to enhance the expressiveness of spike trains.

## 3. Methods

### 3.1. Assigning Weights to Spikes

We apply a specific weight pattern to the spike train to enhance information encoding. The weights decay over time, facilitating rapid transmission of the majority of information, while the minimum weight is constrained to 1 to maintain encoding precision. Specifically, we map the spike train to ANN activation using the following approach:

$$a_i^l \approx \frac{\sum_t \beta^{T-t} S_i^l[t]}{T} = \frac{\sum_{\tau \in \mathbb{F}_i^l} \beta^{T-\tau} \theta^l \delta[t-\tau]}{T} := \bar{r}_i^l \quad (7)$$

where $\beta > 1$ represents the amplification coefficient for weighting. $\beta$ is fixed at 2 in our implementation, resulting in a uniform encoding of $a_i^l$. The approximation symbol indicates that a finite number of timesteps introduces quantization errors. Stöckl & Maass (2021) directly transmits

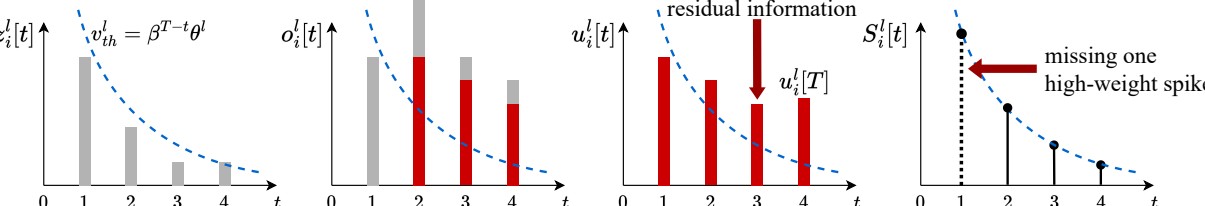

Figure 2. The decaying weight pattern causes information to remain in the membrane potential. *For clarity, weights are directly applied in this illustration.* From left to right, the figure shows the neuron's input, the membrane potential before and after reset, and the emitted spike train. The blue dashed line represents the firing threshold. The residual membrane potential from the previous step is represented by the red bar, which can be identified as the primary contributor to the increase in $u_i^l[T]$. Better control of the residual membrane potential can be achieved by reducing the threshold to promote spike firing.

weighted spike signals across the network, which requires a larger bitwidth for spike representation. Additionally, neurons must adjust their thresholds based on different weights, increasing model complexity. To address this, we integrate the weighting process into the neural computation by incorporating the amplification coefficient $\beta$ into the soft-reset IF model:

$$u_i^l[t] = \beta u_i^l[t-1] + z_i^l[t] - S_i^l[t] \qquad (8)$$

**Proposition 3.1.** *The stepwise weighting process described by Equation (8) is equivalent to directly transmitting weighted spikes. Furthermore, the input and output spike trains satisfy the following relationship:*

$$\bar{r}_i^l = \max\left(\sum_j w_{ij}^l \bar{r}_j^{l-1} + \sum_{t=1}^T \frac{\beta^{T-t} b_i^l}{T} - \frac{u_i^l[T]}{T}, 0\right) \qquad (9)$$

Equation (9) serves as the core equation for ANN-SNN conversion and the detailed derivation can be found in Appendix A.2. By comparing it with Equation (6), we can conclude:

**Corollary 3.2.** *Let $w_{ij}^l = \hat{w}_{ij}^l$ and $b_i^l = \hat{b}_i^l \cdot \frac{T}{\sum_t \beta^{T-t}}$. Assume the input $\bar{r}_j^{l-1}$ encodes $a_j^{l-1}$. To reduce encoding errors for $a_i^l$, the residual membrane potential $u_i^l[T]$ should be minimized.*

Since $T$ typically takes large values in rate coding, previous works often neglect the $u_i^l[T]/T$ term (Rueckauer et al., 2017). However, it is necessary to account for this term when $T$ goes small. Furthermore, we observe that the decaying weight pattern hinders the reduction of $u_i^l[T]$. To address this, we propose the OFC method in the next section to effectively control $u_i^l[T]$.

### 3.2. Reducing Residual Errors

Under a decaying weight pattern, the residual information from previous high-weight inputs often exceeds the encoding capacity of subsequent spikes, leading to an increase

**Algorithm 1** Forward method of the TSA neuron

---

**Input:** input $X$ of shape [TB, C, H, W], length of silent period $P$, spike amplitude $\theta$
**Output:** output spike train $S$ of shape [BT, C, H, W]
Pad $X$ with zeros and reshape to [T+P, B, C, H, W].
Membrane potential $U \leftarrow$ `zeros_like`$(X[0])$
Threshold $v \leftarrow \beta^P \alpha \theta$
/ ∗ silent period ∗ /
**for** $i = 0$ **to** $P - 1$ **do**
    $M \leftarrow \beta M + X[i]$    / ∗ stepwise weighting ∗ /
**end for**
**for** $i = 0$ **to** $T - 1$ **do**
    $M \leftarrow \beta M + X[i+P]$  / ∗ stepwise weighting ∗ /
    / ∗ fire ternary spikes ∗ /
    $S[i] \leftarrow (M \geq v)$.`float()` $- (M \leq -v)$.`float()`
    / ∗ soft reset ∗ /
    $M \leftarrow M - \beta^P \theta S[i]$
**end for**
$S \leftarrow \theta S$

---

in $u_i^l[T]$. We term this phenomenon residual errors, as illustrated in Figure 2. It can be proven that controlling the residual membrane potential at each step is key to minimizing $u_i^l[T]$:

**Proposition 3.3.** $\forall \epsilon > 0$, $u_i^l[T] < \epsilon$ if and only if *for all timestep $\tau \in \{0, 1, \dots, T-1\}$:*

$$u_i^l[\tau] < \frac{\epsilon}{\beta^{T-\tau}} + \frac{1}{\beta^{T-\tau}} \sum_{t=\tau+1}^T \theta^l \beta^{T-t}$$
$$- \frac{1}{\beta^{T-\tau}} \sum_{t=\tau+1}^T \beta^{T-t} z_i^l[t] \qquad (10)$$

Ignoring the coefficient $1/\beta^{T-\tau}$ on the right-hand side, the first summation term represents the maximum value a spike train after $\tau$ can encode, while the second term corresponds to future inputs. These two terms and the given $\epsilon$ impose constraints on the residual membrane potential $u_i^l[\tau]$.

The converted SNN theoretically achieves equivalent performance when $u_i^l[T] < \theta^l$, where only quantization error remains. To achieve this, previous works have proposed delaying the output in the temporal domain: first integrating inputs over *all timesteps*, and then firing spikes continuously (Rueckauer & Liu, 2021; Stöckl & Maass, 2021).

*Remark* 3.4. Proposition 3.3 explains the feasibility of delaying the output: for all $\tau \in \{0, 1, \ldots, T - 1\}$, the maximum encodable value (i.e. the second constraint term) is amplified to $\sum_{t=1}^{T} \theta^l \beta^{T-t}$ as no spikes are fired before $\tau$.

However, this method significantly increases inference latency. Based on Proposition 3.3, we propose constraining $u_i^l[\tau]$ at each step to limit the size of $u_i^l[T]$ with minimal latency overhead.

Specifically, the firing threshold is reduced to $v_{th}^l = \alpha\theta^l$ with $\alpha < 1$. Negative spikes are introduced to correct over-fired information, i.e. $(1 - \alpha)\theta^l$. The negative threshold is set symmetrically to $-\alpha\theta^l$, which triggers a negative spike when $o_i^l[t]$ falls below this value. Given these characteristics, we refer to the neuron model as the TSA neuron and the encoding scheme as the CSS coding scheme.

**Theorem 3.5.** *Let the integrated input $z_i^l[t]$ at each timestep be independent and follow a uniform distribution, $U(0, \theta^l)$. When $\alpha = \frac{1}{2}$, for all $t \in \{1, 2, \ldots, T\}$, we have:*

$$\mathbb{E}(u_i^l[t]) = 0 \quad \text{and} \quad \mathbb{E}(u_i^l[t]^2) \text{ is minimized.}$$

*Remark* 3.6. Considering that $\theta^l$ can reflect the maximum potential input at each timestep, i.e. $\theta^l \approx \max(z_i^l[t])$, the input assumption in Theorem 3.5 is reasonable.

Theorem 3.5 demonstrates that setting $\alpha = 1/2$ most effectively confines the residual membrane potential near zero, with the detailed derivation provided in Appendix A.4. The optimality of $\alpha = 1/2$ has also been experimentally verified in Section 5.4.

Notably, the optimal threshold in Theorem 3.5 also applies to IF neurons, offering a simple yet effective method to reduce encoding errors in rate coding. We present preliminary experimental validation in Appendix B.

However, we find that with decaying weight patterns, $u_i^l[T]$ can exceed $\theta^l$ even with optimal $\alpha$. Therefore, to achieve lossless conversion, we keep a *one-step* output delay. We refer to this as the silent period: neurons integrate input and perform stepwise weighting but are prohibited from firing. The forward method of the TSA neurons is provided in Algorithm 1. After the silent period, the membrane potential is amplified by $\beta$, which leads to a corresponding amplification of both the threshold and the reset amount.

## 3.3. ANN-SNN conversion

The proposed method eliminates the need for directly applying weights to input pixel values, as weighting is integrated during neural computation. Therefore, we adopt the widely used direct coding for input static images (Rueckauer et al., 2017; Li et al., 2021; Hao et al., 2023; Hu et al., 2023): the analog input activations of the first hidden layer are interpreted as constant currents, with spiking outputs starting from this layer.

We replace the ReLU activation function with the TSA neuron to encode the hidden layer activations. Note that since TSA neurons can encode negative activations, additional logic is required to zero out sequences encoding negative values. For $\beta \geq 2$, this logic simplifies to detecting sequences where the first spike is positive, which can be easily implemented.

To determine the spike amplitude $\theta^l$ for each layer, we use the strategy proposed by Rueckauer et al. (2017): after observing ANN activations over a portion of the training set, we calculate the 99.99th percentile $p^l$ of the activation distribution, and then set $\theta^l$ to $p^{l}$[2]. This approach improves the network's robustness to outlier activations. The pseudocode for the conversion process is provided in Appendix C

## 4. Related Works

### 4.1. Spike Coding Schemes

Current mainstream coding schemes in converted SNNs include rate coding and TTFS coding.

Rate coding represents activity by the number of spikes within a time window. Early methods aimed at reducing conversion loss, such as weight normalization (Diehl et al., 2015), threshold rescaling (Sengupta et al., 2019), and soft-reset neurons (Han et al., 2020). More recent work focuses on reducing timesteps by optimizing neuron parameters: Meng et al. (2022) introduced the threshold tuning method, while Bu et al. (2022) proposed optimizing the initial membrane potential. Additionally, recent works have explored quantizing the ANNs before conversion (Bu et al., 2023; Hao et al., 2023; Hu et al., 2023). This approach directly reduces the number of activations that need to be mapped, providing an alternative way to minimize timesteps.

Rueckauer & Liu (2018) were the first to attempt converting an ANN to a TTFS-based SNN, achieving increased sparsity but with significant conversion errors. Stanojevic et al. (2022) showed that exact mapping is feasible. Yang et al. (2023) improved this by using dynamic neuron thresh-

---

[2]More precisely, $\theta^l = p^l \cdot \frac{T}{\sum_{t=1}^{T} \beta^{T-t}}$. Since scaling the spike amplitude of each layer by the same value has no practical effect, we directly set $\theta^l$ to $p^l$ for simplicity.

*Table 2.* Number of timesteps under different neural coding schemes, evaluated on the CIFAR-10 and ImageNet datasets. "No Quant." column indicates whether quantization is required. "No Cal." column marks whether post-conversion fine-tuning is required.

| | Method | No Quant. | No Cal. | Architecture | ANN Acc. | Coding Scheme | Timestep | SNN Acc. |
|---|---|---|---|---|---|---|---|---|
| **CIFAR-10** | OPI (Bu et al., 2022) | ✔ | ✔ | ResNet-20 | 92.74% | rate | 64 | 92.57% |
| | FS-Conversion (Stöckl & Maass, 2021) | ✔ | ✔ | ResNet-20 | 91.58% | FS | 10 | 91.45% |
| | SNN Calibration (Li et al., 2021) | ✔ | ✘ | VGG-16 | 95.72% | rate | 128 | 95.65% |
| | TSC (Han & Roy, 2020) | ✔ | ✔ | VGG-16 | 93.63% | TSC | 512 | 93.57% |
| | TTFS Mapping (Stanojevic et al., 2023) | ✔ | ✘ | VGG-16 | 93.68% | TTFS | 64 | 93.69% |
| | LC-TTFS (Yang et al., 2023) | ✔ | ✔ | VGG-16 | 92.79% | TTFS | 50 | 92.72% |
| | **CSS-SNN**[†] | ✔ | ✔ | ResNet-20 | 93.83% | | 7 | 93.73% |
| | | ✔ | ✔ | VGG-16 | 95.90% | CSS | 8 | 95.92% |
| | | ✔ | ✔ | ResNet-18 | 96.68% | | 6 | 96.62% |
| | QFFS (Li et al., 2022)[†] | ✘ | ✔ | ResNet-18 | 93.12% | rate | 4 | 93.13% |
| | QCFS (Bu et al., 2023) | ✘ | ✔ | ResNet-18 | 96.04% | rate | 16 | 95.92% |
| | COS (Hao et al., 2023) | ✘ | ✔ | ResNet-18 | 95.64% | rate | 4 | 95.46% |
| | Fast-SNN (Hu et al., 2023)[†] | ✘ | ✘ | ResNet-18 | 95.62% | rate | 7 | 95.57% |
| | **CSS-SNN**[†] | ✘ | ✔ | ResNet-18 | 96.32% | CSS | 3 | 96.34% |
| **ImageNet** | TSC (Han & Roy, 2020) | ✔ | ✔ | ResNet-34 | 70.64% | TSC | 4096 | 69.93% |
| | SNN Calibration (Li et al., 2021) | ✔ | ✘ | ResNet-34 | 75.66% | rate | 256 | 74.61% |
| | TSC (Han & Roy, 2020) | ✔ | ✔ | VGG-16 | 73.49% | TSC | 1024 | 73.33% |
| | OPI (Bu et al., 2022) | ✔ | ✔ | VGG-16 | 74.85% | rate | 256 | 74.62% |
| | **CSS-SNN**[†] | ✔ | ✔ | ResNet-34 | 76.42% | CSS | 8 | 76.10% |
| | | ✔ | ✔ | VGG-16 | 75.34% | | 8 | 75.17% |
| | QFFS (Li et al., 2022)[†] | ✘ | ✔ | VGG-16 | 73.08% | rate | 8 | 73.10% |
| | QCFS (Bu et al., 2023) | ✘ | ✔ | VGG-16 | 74.29% | rate | 256 | 74.22% |
| | COS (Hao et al., 2023) | ✘ | ✔ | VGG-16 | 74.19% | rate | 16 | 74.09% |
| | Fast-SNN (Hu et al., 2023)[†] | ✘ | ✘ | VGG-16 | 73.02% | rate | 7 | 72.95% |
| | **CSS-SNN**[†] | ✘ | ✔ | VGG-16 | 74.33% | CSS | 5 | 74.32% |

[†] Utilizing negtive spikes.

old and weight regularization and completed the conversion with 50 timesteps per layer. Han & Roy (2020) introduced the Temporal-Switch-Coding (TSC) scheme, where the time interval between two spikes encodes activation. However, in the above approaches, the improvement in encoding precision relies on a linear increase in the number of timesteps, which limits the performance of the converted SNN at low timesteps.

Using weighted spikes to represent activation values remains an underexplored area. Stöckl & Maass (2021) and Rueckauer & Liu (2021) employed spikes to encode the "1"s in the binary represented activations. However, both approaches require neurons to wait for all input spikes before firing, resulting in high output latency. Kim et al. (2018) sought to reduce encoding errors by repeatedly applying inputs, which requires thousands of timesteps. In contrast, by introducing negative spikes and setting an optimal firing threshold, we achieve fast and accurate computation. Additionally, we propose stepwise weighting during the neuron's decoding

process, enabling a more hardware-friendly architecture.

### 4.2. Negtive Spikes

The role of negative spikes in SNNs has been widely studied in recent works. In rate-based ANN-SNN conversion (Li et al., 2022; Wang et al., 2022; Hu et al., 2023), negative spikes have been utilized to enhance neuron adaptability to input fluctuations, which improves conversion accuracy. Guo et al. (2023) proposed ternary SNNs to increase the expressive capacity of spike sequences and trained them directly using gradient descent. Unlike existing approaches, we intentionally design neuron over-firing and employ negative spikes for correction. We demonstrate that this strategy facilitates efficient computation with weighted spikes.

## 5. Experiments

In this section, we convert ANNs to CSS-coded SNNs and conduct experiments on the CIFAR-10 and ImageNet

*Table 3.* Timesteps for coding vs. Classification accuracy. We conduct experiments with VGG-16 on CIFAR-10 and ImageNet. The results for COS and Fast-SNN are obtained using their open-source code, ensuring *the same pre-conversion ANN accuracy*.

| | Method | Coding Scheme | T=1 | T=2 | T=3 | T=4 | T=5 | T=6 | T=7 | T=8 |
|---|---|---|---|---|---|---|---|---|---|---|
| **CIFAR-10** | COS (Hao et al., 2023) | rate | 89.61% | 94.23% | 95.10% | 95.51% | 95.46% | 95.56% | 95.56% | 95.54% |
| | Fast-SNN (Hu et al., 2023) | rate | 90.13% | 94.50% | 95.39% | 95.53% | 95.60% | 95.59% | 95.64% | 95.63% |
| | **CSS-SNN** | CSS | 89.75% | 95.26% | 95.65% | 95.65% | 95.65% | 95.64% | 95.67% | 95.66% |
| **ImageNet** | COS (Hao et al., 2023) | rate | 0.67% | 37.18% | 66.12% | 71.24% | 72.75% | 73.46% | 73.69% | 73.87% |
| | Fast-SNN (Hu et al., 2023) | rate | 0.61% | 22.52% | 57.84% | 68.71% | 71.81% | 72.75% | 73.26% | 73.54% |
| | **CSS-SNN** | CSS | 0.66% | 66.01% | 73.72% | 74.23% | 74.32% | 74.34% | 74.34% | 74.36% |

datasets. For all experiments, we fix $\beta = 2$ and $\alpha = 1/2$. First, we compare the number of timesteps with other coding schemes. Then, we evaluate the energy consumption. Finally, we conduct ablation studies to validate the effectiveness of the OFC method in minimizing output latency, as well as to verify the optimality of $\alpha = 1/2$

### 5.1. Overall Performance

In Table 2, we compare the number of timesteps required by different coding schemes. The "No Quant." column indicates whether a quantized ANN is required, while the "No Cal." column marks whether post-conversion fine-tuning is applied. Works utilizing negative spikes are denoted with the "†" symbol. Although the accuracy of the ANNs used in each study is provided, conversion loss is a more critical metric for evaluation.

When directly encoding full-precision activations, CSS coding demonstrates nearly lossless conversion with significantly fewer timesteps. For instance, on the ImageNet dataset, Li et al. (Li et al., 2021) reported a conversion loss exceeding 1% for ResNet-34 with 256 timesteps, whereas our method achieved only 0.3% conversion loss with just 8 timesteps. Although FS coding (Stöckl & Maass, 2021) also employs weighted spikes for activation encoding, it incurs higher conversion loss even with more timesteps.

SOTA SNN performance (Hu et al., 2023; Hao et al., 2023) is typically achieved by converting quantized ANNs while still using rate coding. Notably, while rate coding achieves competitive performance with reduced timesteps, it heavily relies on low-bit quantization, which introduces additional training overhead and often compromises accuracy.

Table 3 illustrates the relationship between timesteps and accuracy when encoding quantized activations. The results, based on our reproductions with consistent ANN accuracy, demonstrate that our method integrates seamlessly with quantized ANNs and achieves ANN-level accuracy with fewer timesteps than rate coding, enabling the development of higher-performance SNNs. For example, we achieved 74.23% accuracy on ImageNet with only 4 timesteps.

Furthermore, the CSS coding scheme provides an alternative approach to achieving low-timestep SNNs without aggressive quantization. For example, on CIFAR-10, our method converted a full-precision ResNet-18 with only 6 timesteps and a minimal conversion loss of 0.06%.

### 5.2. Energy Consumption Analysis

In this section, we estimate the energy consumption of our methods[3], with the results summarized in Table 4. To compare with SOTA rate coding, we conduct experiments with quantized ANNs. Notably, the TSA neuron amplifies the membrane potential at each timestep, which can be efficiently implemented via bit-shift operations when $\beta = 2$. We assume shift operations consume the same energy as an Accumulate (AC) operation, providing a conservative overestimation.

TTFS coding (Stanojevic et al., 2023) demonstrates low energy consumption due to its minimized spike count. Although our method does not inherently exhibit sparsity, the reduction in timesteps compensates for this limitation. By further compressing the number of timesteps, our approach achieves a 40% reduction in energy consumption compared to TTFS coding. Additionally, we include Fast-SNN (Hu et al., 2023) and COS (Hao et al., 2023) as strong baselines for (signed) rate coding. The results show that our method outperforms both with a 10% reduction in energy consumption while achieving higher accuracy.

### 5.3. Effect of OFC Method

We conduct an ablation study to demonstrate that the OFC method achieves low-latency computation for weighted spikes. Experiments were conducted using ResNet-34 on ImageNet, where we restored $v_{th}$ to $\theta^l$, removed the negative threshold and increased the length of the silent period.

---

[3]Energy consumption measurements were performed based on https://github.com/iCGY96/syops-counter

*Table 4.* Energy consumption of VGG-16 on CIFAR-10. The results for COS and Fast-SNN are obtained using their open-source code.

| Method | Coding Scheme | Timestep | Accuracy | SyOPs (ACs) | MACs | Energy Consumption |
|---|---|---|---|---|---|---|
| ANN | - | - | 95.61% | 0 | 313.60M | 1.4426mJ |
| TTFS Mapping (Stanojevic et al., 2023)⋆ | TTFS | 64 | 93.53% | 120.53M | 0 | 0.1085mJ |
| COS (Hao et al., 2023) | rate | 4 | 95.51% | 42.28M | 7.34M | 0.0719mJ |
| Fast-SNN (Hu et al., 2023) | rate | 4 | 95.53% | 47.55M | 7.34M | 0.0766mJ |
| **CSS-SNN** | CSS | 3 | 95.65% | 44.39M | 5.51M | 0.0653mJ |

⋆ Stanojevic et al. (2023) reported an average of 0.38 spikes per neuron on VGG-16, which we used to *calculate the SyOPs* and estimate the energy consumption. The calculation method is the same as the one used in the code. See Appendix D for computation details.

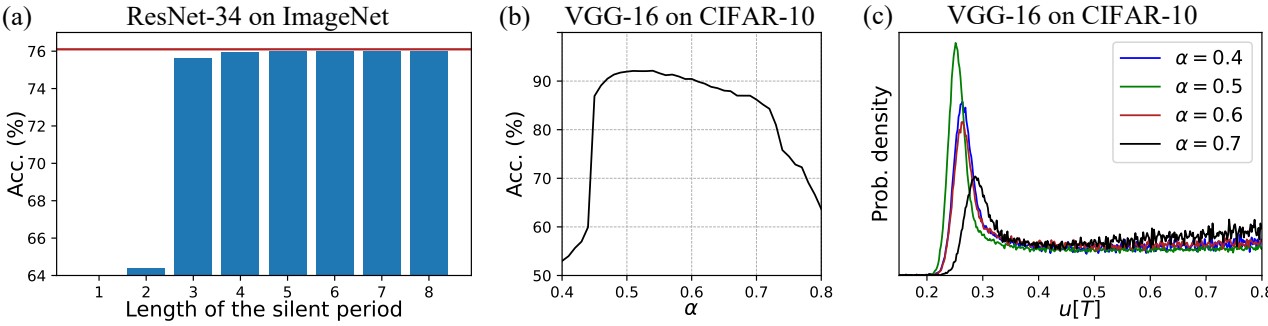

*Figure 3.* (a) Relationship between silent period length and accuracy. The blue bars represent the results obtained relying solely on delaying output. The red baseline indicates the results obtained after applying the OFC method, with a silent period of 1 step. Without using the OFC method, a longer silent period is required to achieve accurate weighted spike computation, which increases the network's output latency. (b) Accuracy variations with different $\alpha$ values. (c) Distribution of residual membrane potential for various $\alpha$ values.

Assuming the $P$-step silent period for each layer is processed in a pipelined manner, its contribution to the output latency of an $L$-layer network is $P \times L$.

The experimental results shown in Figure 3 (a) indicate that a silent period of five steps is required to match the performance achieved by OFC at $P = 1$, which corresponds to a 136-step latency reduction for ResNet-34. Compared to previous works where $P = T$ (Stöckl & Maass, 2021; Rueckauer & Liu, 2021), with $T$ denoting timesteps for encoding, the OFC method reduces output latency by a factor of $T$.

### 5.4. Threshold Setting

To validate the rationale for our threshold setting, we varied $\alpha$ within $[0.4, 0.8]$ with a precision of $0.01$ and evaluated the classification accuracy. Experiments were conducted on CIFAR-10 using VGG-16. To better highlight the impact of threshold adjustment, we disabled the silent period.

The experimental results, shown in Figure 3 (b), demonstrate that network performance peaks when $\alpha$ is around $0.5$. In Figure 3 (c), we plot the distribution of the average residual membrane potential $u[T]$ across all neurons

for different thresholds, revealing that $\alpha = 0.5$ most effectively constrains $u[T]$. We further validated this across VGG architectures of varying depths. The results provided in Appendix E consistently support this conclusion, which demonstrate the robustness of the optimized $\alpha$ value.

## 6. Conclusion

In this work, we explore the use of weighted spike trains to efficiently encode ANN activations. We propose step-wise weighting during neural computation, resulting in a simpler neuron model. Communication between neurons does not require additional bit width for weight information. We introduce the OFC method to enable fast and accurate computation with weighted spikes. Experimental results demonstrate that the CSS coding scheme significantly reduces the number of timesteps to encode activations while maintaining minimal conversion loss. This approach offers the potential to enable high-performance SNNs directly from full-precision ANNs, reducing the reliance on quantization in current mainstream conversion frameworks. Furthermore, the CSS coding scheme achieves lower energy consumption in the converted SNNs.

## Impact Statement

This paper presents work whose goal is to advance the field of Machine Learning. There are many potential societal consequences of our work, none which we feel must be specifically highlighted here.

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

## A. Mathematical Proofs

### A.1. Proof of Equation (5)

$$\frac{\sum_t S_i^l[t]}{T} = \sum_j w_{ij}^l \frac{\sum_t S_j^{l-1}[t]}{T} + \sum_{t=1}^{T} \frac{b_i^l}{T} - \frac{u_i^l[T]}{T}$$

*Proof.* Starting with the initial condition $u_i^l[0] = 0$ and Equation (3), we can write:

$$u_i^l[1] = z_i^l[1] - S_i^l[1] \tag{A1}$$

Next, we derive the expression for $u_i^l[2]$ by substitute the above into Equation (3):

$$u_i^l[2] = z_i^l[1] - S_i^l[1] + z_i^l[2] - S_i^l[2] \tag{A2}$$

We can generalize this process to iteratively compute the membrane potential up to $t = T$:

$$u_i^l[T] = \sum_{t=1}^{T} z_i^l[t] - S_i^l[t] \tag{A3}$$

Substituting $z_i^l[t]$ with Equation (4) and rearranging the terms, we get:

$$S_i^l[t] = \sum_{t=1}^{T} \sum_j w_{ij}^l S_j^{l-1}[t] + b_i^l - u_i^l[T] \tag{A4}$$

Exchange the order of summation and devide both sides by $T$, we can write:

$$\frac{\sum_t S_i^l[t]}{T} = \sum_j w_{ij}^l \frac{\sum_t S_j^{l-1}[t]}{T} + \sum_{t=1}^{T} \frac{b_i^l}{T} - \frac{u_i^l[T]}{T} \tag{A5}$$

$\square$

### A.2. Proof of Proposition 3.1

**Proposition A.1.** *The stepwise weighting process described by Equation (8) is equivalent to directly transmitting weighted spikes. Furthermore, the input and output spike trains satisfy the following relationship:*

$$\bar{r}_i^l = \max\left(\sum_j w_{ij}^l \bar{r}_j^{l-1} + \sum_{t=1}^{T} \frac{\beta^{T-t} b_i^l}{T} - \frac{u_i^l[T]}{T}, 0\right)$$

*Proof.* Following a similar derivation as in Appendix A.1, we can write:

$$u_i^l[T] = \sum_{t=1}^{T} \beta^{T-t}(z_i^l[t] - S_i^l[t]) \tag{A6}$$

Substituting $z_i^l[t]$ with Equation (4) and reorganizing the terms, we get:

$$u_i^l[T] = \sum_j w_{ij}^l \sum_{t=1}^{T} \beta^{T-t} S_j^{l-1}[t] + \sum_{t=1}^{T} \beta^{T-t} b_i^l - \sum_{t=1}^{T} \beta^{T-t} S_i^l[t] \tag{A7}$$

Based on Equation (A7), we can conclude that stepwise weighting is equivalent to directly receiving and firing weighted spikes. Reorganize the terms and devide both sides by $T$, we have:

$$\frac{\sum_{t=1}^{T} \beta^{T-t} S_i^l[t]}{T} = \sum_j w_{ij}^l \frac{\sum_{t=1}^{T} \beta^{T-t} S_j^{l-1}[t]}{T} + \frac{\sum_{t=1}^{T} \beta^{T-t} b_i^l}{T} - \frac{u_i^l[T]}{T} \tag{A8}$$

Following the definition of $\bar{r}_i^l$ in Equation (7) and note that $\frac{\sum_t \beta^{T-t} S_i^l[t]}{T} \geq 0^4$, we can write:

$$\bar{r}_i^l = \max\left(\sum_j w_{ij}^l \bar{r}_j^{l-1} + \sum_{t=1}^{T} \frac{\beta^{T-t} b_i^l}{T} - \frac{u_i^l[T]}{T}, 0\right) \tag{A9}$$

$\square$

### A.3. Proof of Proposition 3.3

**Proposition A.2.** $\forall \epsilon > 0$, $u_i^l[T] < \epsilon$ if and only if *for all timestep $\tau \in \{0, 1, \ldots, T-1\}$:*

$$u_i^l[\tau] < \frac{\epsilon}{\beta^{T-\tau}} + \frac{1}{\beta^{T-\tau}} \sum_{t=\tau+1}^{T} \theta^l \beta^{T-t} - \frac{1}{\beta^{T-\tau}} \sum_{t=\tau+1}^{T} \beta^{T-t} z_i^l[t]$$

*Proof.* We first prove the forward direction. Given that $u_i^l[T] < \epsilon$, we can express it using Equation (8) as follows:

$$\beta u_i^l[T-1] + z_i^l[T] < \epsilon + S_i^l[T] \leq \epsilon + \theta^l \tag{A10}$$

Substitute $u_i^l[T-1]$ with Equation (8) and we can write:

$$\beta^2 u_i^l[T-2] + \beta z_i^l[T-1] + z_i^l[T] < \epsilon + \theta^l + \beta \theta^l \tag{A11}$$

For all timestep $\tau \in \{1, 2, \ldots, T\}$, the above process can be repeated until we obtain an equation involving $u_i^l[\tau]$:

$$\beta^{T-\tau} u_i^l[\tau] + \sum_{t=\tau+1}^{T} \beta^{T-t} z_i^l[t] < \epsilon + \sum_{t=\tau+1}^{T} \theta^l \beta^{T-t} \tag{A12}$$

Reorganizing Equation (A12) and dividing both sides by $\beta^{T-\tau}$, the validity of the forward reasoning is established.

Next, we proceed to prove the backward direction. For any $\tau \in \{0, 1, \ldots, T-1\}$, by iteratively updating the membrane potential using Equation (8) from $t = \tau + 1$ until $t = T$ and then substituting $z_i^l[t]$ with Equation (4), we can get:

$$u_i^l[T] = \beta^{T-\tau} u_i^l[\tau] + \sum_{t=\tau+1}^{T} \beta^{T-t} z_i^l[t] - \sum_{t=\tau+1}^{T} \beta^{T-t} S_i^l[t] \tag{A13}$$

Note that $\sum_t \beta^{T-t} S_i^l[t] \leq \sum_t \theta^l \beta^{T-t}$. Then we can write:

$$u_i^l[T] \leq \beta^{T-\tau} u_i^l[\tau] + \sum_{t=\tau+1}^{T} \beta^{T-t} z_i^l[t] - \sum_{t=\tau+1}^{T} \theta^l \beta^{T-t} < \epsilon \tag{A14}$$

$\square$

### A.4. Proof of Theorem 3.5

**Theorem A.3.** *Let the integrated input $z_i^l[t]$ at each timestep be independent and follow a uniform distribution, $U(0, \theta^l)$. When $\alpha = \frac{1}{2}$, for all $t \in \{1, 2, \ldots, T\}$, we have:*

$$\mathbb{E}(u_i^l[t]) = 0 \quad \text{and} \quad \mathbb{E}(u_i^l[t]^2) \text{ is minimized.}$$

*Proof.* Let $p(z)$ denote the probability density function of the input $z_i^l[t]$. Define $k_i^l[t] = \beta u_i^l[t]$. For simplicity, we will

---

[4]For the ternary neuron model, we ensure this condition by filtering out spike trains encoding negative values, as detailed in Section 3.3.

drop both neuron and layer index and denote $z[t]$ and $k[t]$ as $z$ and $k$, respectively. According to Equation (8), we have:

$$\mathbb{E}(u[t+1]) = \mathbb{E}_k \left( \int_{-\infty}^{\infty} (z+k)p(z)\mathrm{d}z - \theta^l \int_{\alpha\theta^l - k}^{\infty} p(z)\mathrm{d}z \right)$$

$$= \mathbb{E}_k \left( E(z) + k - \theta^l \int_{\alpha\theta^l - k}^{\infty} p(z)\mathrm{d}z \right) \tag{A15}$$

$$= \mathbb{E}(z) + \mathbb{E}(k) - \theta^l \mathbb{E}_k \left( \int_{\alpha\theta^l - k}^{\infty} p(z)\mathrm{d}z \right)$$

Let $q(k)$ denote the probability density function of $k$. We can write:

$$\mathbb{E}_k \left( \int_{\alpha\theta^l - k}^{\infty} p(z)\mathrm{d}z \right) = \int_{-\infty}^{(\alpha-1)\theta^l} q(k) \int_{\alpha\theta^l - k}^{\infty} p(z)\mathrm{d}z\mathrm{d}k$$

$$+ \int_{(\alpha-1)\theta^l}^{\alpha\theta^l} q(k) \int_{\alpha\theta^l - k}^{\infty} p(z)\mathrm{d}z\mathrm{d}k + \int_{\alpha\theta^l}^{\infty} q(k) \int_{\alpha\theta^l - k}^{\infty} p(z)\mathrm{d}z\mathrm{d}k \tag{A16}$$

$$= \int_{(\alpha-1)\theta^l}^{\alpha\theta^l} q(k) \int_{\alpha\theta^l - k}^{\infty} p(z)\mathrm{d}z\mathrm{d}k + \int_{\alpha\theta^l}^{\infty} q(k)\mathrm{d}k$$

$$= \mathbb{E}_k \left( F_z(\infty) - F_z(\alpha\theta^l - k) \right) + F_k(\infty) - F_k(\alpha\theta^l)$$

where $F_z(\cdot)$ and $F_k(\cdot)$ denote the cumulative distribution functions of $z$ and $k$, respectively. Note that $Z \sim U(0, \theta^l)$, so $F_z(\cdot)$ is linear. We further assume that $k$ is almost constrained within the threshold, i.e., $F_k(\alpha\theta^l) \approx 1$. Therefore, we have:

$$\mathbb{E}_k \left( \int_{\alpha\theta^l - k}^{\infty} p(z)\mathrm{d}z \right) \approx 1 - F_z(\alpha\theta^l - \mathbb{E}(k)) \tag{A17}$$

and

$$\mathbb{E}(u[t+1]) = \mathbb{E}(z) + \mathbb{E}(k) - \theta^l \left( 1 - F_z(\alpha\theta^l - \mathbb{E}(k)) \right) \tag{A18}$$

Note that $k[0] = 0$ and $F_z\left(\frac{1}{2}\theta^l\right) = \frac{1}{2}$. When $\alpha = \frac{1}{2}$, we have:

$$\mathbb{E}(u[1]) = \mathbb{E}(z) - \theta^l \left( 1 - F_z(\frac{1}{2}\theta^l) \right) = 0 = \mathbb{E}(k[1]) \tag{11}$$

By repeatedly applying Equation (A18), we can conclude that for all $t \in \{1, 2, \dots, T\}$, $\mathbb{E}(u_i^l[t]) = 0$.

Similarily, we can write:

$$\mathbb{E}(u[t+1]^2) = \mathbb{E}_k \left( \int_{-\infty}^{\alpha\theta^l - k} (z+k)^2 p(z)\mathrm{d}z + \int_{\alpha\theta^l - k}^{\infty} (z+k-\theta^l)^2 p(z)\mathrm{d}z \right)$$

$$= \mathbb{E}_k \left( \int_{-\infty}^{\alpha\theta^l} z^2 p(z-k)\mathrm{d}z + \int_{\alpha\theta^l}^{\infty} (z-\theta^l)^2 p(z-k)\mathrm{d}z \right) \tag{A19}$$

$$= \mathbb{E}_k \left( \int_{-\infty}^{\infty} z^2 p(z-k)\mathrm{d}z - \theta^l \int_{\alpha\theta^l}^{\infty} (2z-\theta^l)p(z-k)\mathrm{d}z \right)$$

Taking the derivative of the above equation with respect to $\alpha$ and exchanging the order of differentiation and integration (i.e., $\mathbb{E}(\cdot)$), we obtain:

$$\frac{\partial \mathbb{E}(u[t+1]^2)}{\partial \alpha} = -\mathbb{E}_k \left( \frac{\partial}{\partial \alpha} \theta^l \int_{\alpha\theta^l}^{\infty} (2z-\theta^l)p(z-k)\mathrm{d}z \right)$$

$$= \mathbb{E}_k \left( \frac{\partial}{\partial \alpha} \theta^l \int_{\infty}^{\alpha\theta^l} (2z-\theta^l)p(z-k)\mathrm{d}z \right) \tag{A20}$$

$$= (2\alpha - 1)(\theta^l)^3 \mathbb{E}_k \left( p(\alpha\theta^l - k) \right)$$

Note that $\mathbb{E}_k \left( p(\alpha\theta^l - k) \right) > 0$. The derivative is negative when $\alpha < \frac{1}{2}$ and positive when $\alpha > \frac{1}{2}$. Therefore, when $\alpha = \frac{1}{2}$, $\mathbb{E}(u_i^l[t]^2)$ is minimized for all $t \in \{1, 2, \dots, T\}$. $\square$

## B. Applying optimal threshold to IF neurons

*Table 5.* Timesteps vs. Accuracy under rate coding with and w/o optimal threshold.

| Use Opt. threshold | T=4 | T=8 | T=16 | T=24 | T=32 | T=64 | T=128 |
|---|---|---|---|---|---|---|---|
| ✘ | 10.00% | 10.00% | 55.52% | 91.05% | 94.02% | 95.32% | 95.80% |
| ✔ | 33.55% | 77.80% | 90.40% | 93.34% | 94.40% | 95.55% | 95.83% |

In this section, we use VGG-16 to conduct experiments on CIFAR-10 and compare two threshold settings for IF neurons: $v_{th}^l = \frac{1}{2}\theta^l$ and $v_{th}^l = \theta^l$. Notably, since each spike carries equal weight, the over-fired information can be treated as quantization error. The experimental results in Table 5 demonstrate that setting $v_{th}^l = \frac{1}{2}\theta^l$ effectively reduces encoding errors in rate coding. When $T$ is relatively small (e.g., $T \le 16$), the last term in Equation (5), i.e. the residual error, can no longer be ignored. In this case, the optimal threshold effectively controls the encoding error, leading to a significant performance improvement.

## C. Algorithm for ANN-SNN conversion

---
**Algorithm 2** Algorithm for ANN-SNN conversion under CSS coding.

---
**Input:** ANN model $f_A(\hat{W}, \hat{b})$, number of timesteps $T$, number of batches $B$, number of layers $L$.
**Output:** SNN models $f_S(W, b)$
/∗ determine $\theta^1$ ∗/
**for** $l = 0$ **to** $L - 1$ **do**
  $\bar{p}^l \leftarrow 0$
  **for** $n = 0$ **to** $B - 1$ **do**
    $p^l \leftarrow$ 99.99th percentile of $a^l$ distribution
    /∗ average ∗/
    $\bar{p}^l \leftarrow \bar{p}^l + p^l/B$
  **end for**
  $\theta^l \leftarrow \bar{p}^l$
**end for**
**for** $l = 0$ **to** $L - 1$ **do**
  /∗ copy weight and bias ∗/
  $W^l \leftarrow \hat{W}^l$
  $b^l \leftarrow \hat{b}^l$
  Replace ReLU activation with TSA.
**end for**

---

## D. Energy Consumption Analysis

The energy consumption of inferring a single image can be estimated by the following equation:

$$E = T \times E_{AC} \times \sum_l FLOPs(l) \times R(l) \tag{A21}$$

where $T$ denotes the number of timesteps, $E_{AC}$ denotes the energy consumption of an AC operation, $R(l)$ denotes the firing rate of the $l$-th layer. $FLOPs(l)$ denotes the number of floating-point operations in $l$-th layer:

$$FLOPs(l) = \begin{cases} (K^l)^2 \times W^l \times H^l \times C_{in}^l \times C_{out}^l, & Conv\ layer \\ C_{in}^l \times C_{out}^l, & Linear\ layer \end{cases} \tag{A22}$$

Stanojevic et al. (2023) reported an average of 0.38 spikes per neuron on VGG-16. Assuming similar neuronal activity levels across layers, we estimate energy consumption for TTFS coding with the following equation:

$$E = 0.38 \times E_{AC} \times \sum_{l} FLOPs(l) \tag{A23}$$

# E. Additional Results for Section 5.4

**Results from VGG-11:**

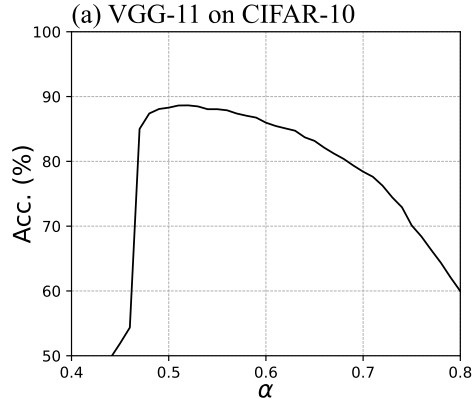 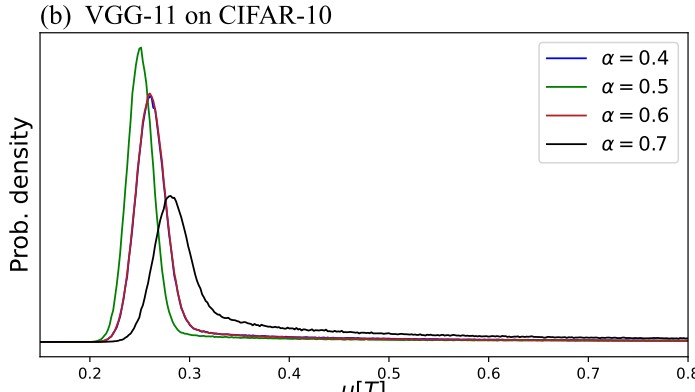

**Results from VGG-13:**

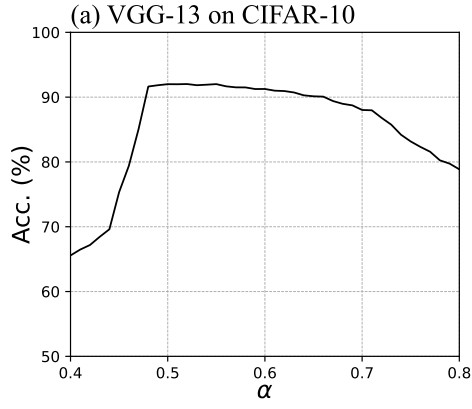 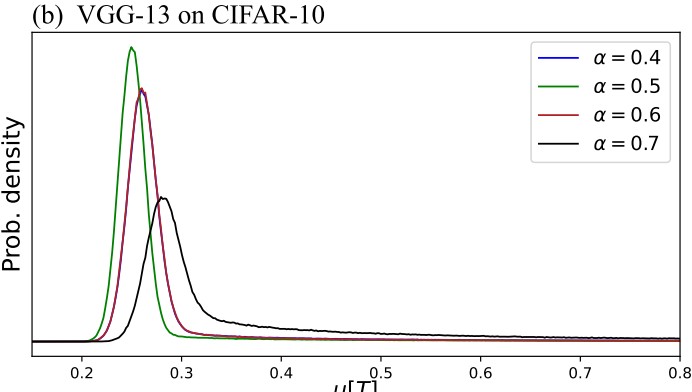

