# OpenReview forum: "Canonic Signed Spike Coding for Efficient Spiking Neural Networks"
_ICML.cc/2025/Conference — Submitted to ICML 2025_

### Official Review · Reviewer_pam4 · 2025-03-11

**Overall Recommendation:** 2

**Summary:**

The paper aims to improve the conversion of Artificial Neural Networks (ANNs) to Spiking Neural Networks (SNNs) by developing a more efficient spike coding scheme, which has improved encoding capacity and reduced computational overhead.

**Claims And Evidence:**

Based on a careful review, the claims in the paper are generally well-supported by evidence.

**Essential References Not Discussed:**

No.

**Experimental Designs Or Analyses:**

The experimental designs are sound, validating the proposed Canonic Signed Spike coding scheme effectively. Experimental results demonstrate their method's performance and advantages.

**Methods And Evaluation Criteria:**

The proposed methods and evaluation criteria align with the research problem that reduces the conversion loss between ANN and SNN.

**Other Comments Or Suggestions:**

It is recommended that the author refer to Fast-SNN[1] and Spike-Zip-TF[2] for experimental design.
[1] Fast-snn: Fast spiking neural network by converting quantized ann
[2] SpikeZIP-TF: Conversion is All You Need for Transformer-based SNN

**Other Strengths And Weaknesses:**

Strengths:
1. The authors develop a spike coding scheme called CSS that has improved encoding capacity and reduced computational overhead.
2. The paper's theoretical proofs are mathematically rigorous and support the proposed CSS coding scheme.
3. Experimental results demonstrate their method's performance.

Weaknesses & Questions:
1. Can the paper's method be extended to broader network architectures, such as Spiking Transformers that contain the attention mechanism?
2. Why did the authors only validate their method on simple image classification tasks? Noteworthy, the state-of-the-art methods in ANN-SNN conversion like Fast-SNN[1] and Spike-Zip-TF[2] have verified their methods across multiple tasks, such as object detection, semantic segmentation, and natural language understanding.
3. The SyOPs and energy efficiency claims in Table 4 require additional re-examination. Why is the energy consumption calculated by the rate-based and the CSS-based method lower than that of the single-spike TTFS method?
4. Can the proposed method support conversion on neuromorphic datasets?
5. Why were the authors not compared with the most recent state-of-the-art ANN-to-SNN conversion algorithms[2,3,4]?
Refs:
[1] Fast-snn: Fast spiking neural network by converting quantized ann
[2] SpikeZIP-TF: Conversion is All You Need for Transformer-based SNN
[3] Towards High-performance Spiking Transformers from ANN to SNN Conversion
[4] A universal ANN-to-SNN framework for achieving high accuracy and low latency deep Spiking Neural Networks

**Questions For Authors:**

See Weaknesses & Questions.

**Relation To Broader Scientific Literature:**

The paper's key contributions are within the ANN-SNN conversion learning algorithm. More specifically, the authors extend work by Li et al. (2022) and Wang et al. (2022) that uses negative spikes, and then introduces a more systematic approach to negative spike correction. Compared to previous methods, the proposed methods provide more efficient information encoding.

**Theoretical Claims:**

The paper's theoretical proofs are mathematically rigorous and provide support for the proposed Canonic Signed Spike (CSS) coding scheme.

---

> ### Author Rebuttal · Authors · 2025-04-01
>
> Thank you for your thorough review. Below, we address some key points of your concerns.
>
> ---
>
> First, we would like to emphasize that our core contribution lies in the innovation of the ___encoding method___. Our work is not a continuation of Li et al. (2022) and Wang et al. (2022) because our core focus is on weighting the spikes, whereas they still rely on rate coding. Although both approaches use negative spikes, their roles are different, as we explained in Section 4.2.
>
> By leveraging spike weighting, we enhance the encoding capacity of spike sequences. Compared to mainstream rate coding or TTFS coding, our approach significantly reduces the required timesteps. __Our method follows the existing ANN-SNN conversion framework, making it straightforward to transition from rate coding to CSS coding__. Since the core objective of ANN-SNN conversion is to accurately represent ANN activations using spike sequences, our approach is not limited to specific network architectures or tasks but rather __provides a broadly applicable optimization__.
>
> Our experimental design aims to verify the accuracy of the encoding and the effectiveness of the TSA design. We choose image classification as the primary task because it is the most common benchmark in SNN research. We adopt CNN architectures as they are still the most widely used in ANN-SNN conversion. Additionally, we compare with CNNs of _the same structure_ to highlight the impact of the encoding method, thereby better demonstrating the value of our work. To address your concerns regarding applicability, we have further extended our method to ViTs and object detection tasks, with preliminary results provided below.
>
> ### Extended Experiments
>
> #### Conversion of Transformer architectures
>
> To demonstrate that CSS can also encode activations in Transformers, we converted ViT-S and ViT-B for the ImageNet classification task. We used the pre-trained weights provided in SpikeZIP-TF [1], where "32Level" denotes the quantization precision. As shown, our encoding scheme significantly reduces the required timesteps under the same weights. Additionally, we provide the actual runtime for a more intuitive comparison.
>
> Method|Architecture|Param|Encoding Scheme|Timestep|Acc.|Runtime|
> :-|:-|:-|:-|:-|:-|:-
> SpikeZIP-TF|ViT-S-32Level|22.05M|rate|64|81.45%|3492.62s
> CSS-SNN|ViT-S-32Level|22.05M|CSS|6|81.51%|325.55s
>
>
> #### Object detection tasks
>
> To demonstrate that CSS can be applied to object detection tasks, we conducted experiments on the VOC2007 dataset using the same architecture and weights as in Fast-SNN [2]. The results are shown in the table below. As observed, our method not only reduces the required timesteps but also significantly lowers the conversion loss.
>
> Method|Architecture|ANN mAP|Encoding Scheme|Timestep|SNN mAP|
> :-|:-|:-|:-|:-|:-|
> Fast-SNN|YOLOv2(ResNet-34-4b)|76.16|rate|15|76.05|
> CSS-SNN|YOLOv2(ResNet-34-4b)|76.16|CSS|4|76.18|
> Fast-SNN|YOLOv2(ResNet-34-3b)|75.27|rate|7|73.43|
> CSS-SNN|YOLOv2(ResNet-34-3b)|75.27|CSS|3|75.20|
>
> In addition, we have also included the experiment on the neuromorphic dataset in our response to Reviewer cfNu, which you may find useful for reference.
>
> We would like to emphasize once again that __our core contribution lies in the nonlinear encoding scheme, which has broad applicability__. For existing rate-based ANN-SNN frameworks, one only needs to replace the IF neurons with TSA neurons, as we have done in the experiments above.
>
> [1] SpikeZIP-TF: Conversion is All You Need for Transformer-based SNN
> [2] Fast-SNN: Fast Spiking Neural Network by Converting Quantized ANN
>
> ### Energy Estimation
>
> Our energy estimation is based on an open-source codebase, and we have carefully re-examined the code without identifying any issues. Although it may seem counterintuitive, it is entirely plausible that CSS exhibits lower energy consumption than TTFS. First, CSS operates with _only three timesteps_, meaning each neuron can fire at most three spikes. Second, the CSS encoding scheme applies _a more aggressive quantization to ANN activations_—many small activations in the ANN are encoded as zero. In contrast, TTFS _utilizes more timesteps for fine-grained quantization_, encoding a greater number of "less important" activations. As a result, __although TSA neurons may fire multiple spikes, significantly fewer neurons are activated in CSS coding__. This combined effect leads to CSS achieving lower overall energy consumption.
>
> ---
>
> If you have any further questions, we would be happy to address them.

---

> > ### Comment · Reviewer_pam4 · 2025-04-02
> >
> > Thank you for your reply. I am very grateful to the author for conducting additional experiments to answer my questions, and some of my questions have been resolved, so I will modify my rating. However, I still have the following concerns/questions.
> >
> > 1. Can the proposed method support neuromorphic datasets?
> >
> > 2. The authors have effectively demonstrated the feasibility of their approach in visual processing tasks. As they note, "the core contribution lies in the nonlinear encoding scheme, which has broad applicability. For existing rate-based ANN-SNN frameworks, one only needs to replace the IF neurons with TSA neurons." However, it remains unclear whether this method could be successfully applied to text-based tasks such as NLP or NLU. Furthermore, the potential applicability to speech processing tasks, which inherently contain richer temporal information structures, warrants investigation. How might the proposed encoding scheme perform when extended to these different tasks?
> >
> > 3. The authors state: "our approach follows a nonlinear accumulation process. This key difference allows us to accumulate information more quickly, thereby reducing the required timesteps." Does the proposed method introduce floating-point multiplication operations? Does this compromise the important spike-driven advantages of SNNs?
> >
> > 4. Although TTFS uses more time steps for fine-grained quantization, it emits at most one spike across all time steps. With the same network structure, the number of spikes emitted by the TTFS encoding should be less than the method proposed in this paper. Additionally, how should we understand the author's explanation of " TTFS encodes more 'less important' activations"?

---

> > > ### Author Response · Authors · 2025-04-05
> > >
> > > Thank you for raising your score and providing further feedback! We are happy to address your concerns.
> > >
> > > ### Additional Experiments
> > >
> > > #### Neuromorphic Datasets
> > >
> > > We have already presented results on the _DVS128Gesture_ dataset in our response to Reviewer cfNu. Here, we further include experiments with ResNet-18 on _CIFAR10-DVS_ and _N-Caltech101_ datasets. We implemented a simple rate coding scheme as the baseline, and the results are presented in the table below. The results demonstrate that our method is __fully compatible with neuromorphic datasets__, significantly reduces the number of timesteps, and further mitigates the conversion loss.
> > >
> > > Method|ANN Acc.|Dataset|Coding Scheme|T|SNN Acc.
> > > :-|:-|:-|:-|:-|:-
> > > -|90.94%|DVS128Gesture|rate|128|90.56%
> > > CSS-SNN|90.94%|DVS128Gesture|CSS|6|90.89%
> > > -|83.03%|N-Caltech101|rate|128|82.51%
> > > CSS-SNN|83.03%|N-Caltech101|CSS|8|82.76%
> > > -|78.35%|CIFAR10-DVS|rate|256|77.87%
> > > CSS-SNN|78.35%|CIFAR10-DVS|CSS|8|78.15%
> > >
> > > #### Natural Language Processing
> > >
> > > We have already demonstrated the effectiveness of our method on the Transformer architecture, __making its application to NLP tasks a natural extension__. We conducted experiments using the RoBERTa model on the _IMDB Movie Review_ and _SST-2_ datasets, using the pretrained ANN provided in SpikeZIP-TF for conversion. The results are presented in the table below, demonstrating the effectiveness of our method on NLP tasks. We additionally report the runtime, which clearly highlights the efficiency of our approach.
> > >
> > > Method|Arch|Dataset|Coding Scheme|T|Acc.|Runtime|
> > > :-|:-|:-|:-|:-|:-|:-
> > > SpikeZIP-TF|RoBERTa-B-32Lv|SST-2|rate|64|92.32%|169.45s
> > > CSS-SNN|RoBERTa-B-32Lv|SST-2|CSS|5|92.32%|19.68s
> > > SpikeZIP-TF|RoBERTa-B-32Lv|IMDB-MR|rate|64|81.30%|4964.80s
> > > CSS-SNN|RoBERTa-B-32Lv|IMDB-MR|CSS|5|81.36%|489.51s
> > >
> > > #### Audio Classification
> > >
> > > Our method can also be applied to audio processing. We conduct audio classification tasks using ResNet-18 on the _GTZAN_ and _ESC-50_ datasets. The results, shown in the table below, further demonstrate the strong applicability of our method.
> > >
> > > Method|ANN Acc.|Dataset|Coding Scheme|T|SNN Acc.
> > > :-|:-|:-|:-|:-|:-
> > > -|90.62%|GTZAN|rate|256|89.54%
> > > CSS-SNN|90.62%|GTZAN|CSS|8|90.28%
> > > -|75.15%|ESC-50|rate|256|74.97%
> > > CSS-SNN|75.15%|ESC-50|CSS|8|75.00%
> > >
> > >
> > > We have already demonstrated the applicability of our design across visual, textual, and auditory tasks, as you suggested. We sincerely appreciate your constructive suggestions and believe these additional results further enrich our work. However, we hope you understand that it is impractical to enumerate and evaluate our method on every possible task.
> > >
> > > We would like to reiterate that __our method is not limited to specific models or tasks__. Instead, it __introduces a general innovation in encoding__. Our proposed CSS coding significantly reduces the number of timesteps while preserving the simplicity of the rate-based conversion process. Wherever ANN-to-SNN conversion is applicable, our method can be readily adopted. This makes our approach a substantial contribution to the SNN community.
> > >
> > > ### Further Clarifications
> > >
> > > #### Efficient Implementation of Spike Weighting
> > >
> > > In our method, spike weights are applied by _doubling the membrane potential at each timestep_, which corresponds to a left shift in hardware. Since the shift amount is fixed, _it can be implemented purely through wiring_, eliminating the need for shift registers. __This design introduces no floating-point operations, preserves the spike-driven nature of SNNs, and incurs negligible hardware cost__. For a more intuitive understanding, we provide an illustration of the reference design at [this URL](https://anonymous.4open.science/r/ICML-2025-Rebuttal-0DE7/README.md). You may also refer to our response to Reviewer MkoZ for more detailed information.
> > >
> > > #### Reduced Spike Count
> > >
> > > To address your follow-up question, we offer a more detailed breakdown of the spike counts. Suppose the activation value lies within $[0, x_p]$:
> > >
> > > - For TTFS coding [2] with $T = 64$, _activations in $[\frac{1}{64}x_p, x_p]$_ will be encoded as exactly _one spike_;
> > > - For CSS coding with $T = 3$, _activations in $[\frac{1}{8}x_p, x_p]$_ are encoded into _a spike sequence_.
> > >
> > > Considering the typical activation distribution in ANNs—where a large proportion of activations are close to zero—TTFS ends up encoding more values. Also note that CSS uses only three timesteps, requiring just around 1.5 spikes per activation. We visualize the activation distribution and report the average spike counts for encoding in each layer at [this URL](https://anonymous.4open.science/r/ICML-2025-Rebuttal-Act-Dist-C538/README.md).
> > >
> > > ---
> > >
> > > Finally, we believe our method introduces a meaningful innovation and delivers strong practical effectiveness, contributing to the advancement of ANN-SNN conversion. We sincerely appreciate the time and effort you have dedicated to reviewing our work!

---

### Official Review · Reviewer_MkoZ · 2025-03-11

**Overall Recommendation:** 2

**Summary:**

The paper proposes the Canonic Signed Spike (CSS) coding scheme, which enhances encoding capacity while maintaining network simplicity. Additionally, the Over-Fire-and-Correct method is introduced to enable efficient computation. The primary contribution lies in minimizing conversion loss when transforming artificial neural networks (ANNs) into spiking neural networks (SNNs).

**Claims And Evidence:**

The paper claims to achieve minimal conversion loss from ANN to SNN while preserving computational efficiency. However, while the proposed approach is promising, it shares similarities with existing methods, particularly the work of Stöckl & Maass (2021). Despite this, the proposed method does not appear to surpass existing techniques in terms of flexibility.

**Essential References Not Discussed:**

The references used in the paper are well-structured and appropriate for the study.

**Experimental Designs Or Analyses:**

The experimental evaluation conducted on CIFAR-10 and ImageNet is reasonable and aligns with the standard benchmarks used in SNN research.

**Methods And Evaluation Criteria:**

The validation methods used in the paper are correct. The authors apply Leaky Integrate-and-Fire (LIF) neurons for SNN conversion and introduce modulation mechanisms to ensure precise transformation. The methodology is well-grounded in established conversion techniques.

**Other Comments Or Suggestions:**

None.

**Other Strengths And Weaknesses:**

The proposed method does not require model quantization, which reduces training requirements and enhances deployment efficiency.

**Questions For Authors:**

Can the proposed method be extended to support activation functions beyond ReLU, similar to the approach in Stöckl & Maass (2021)?

The paper claims that the algorithm is hardware-friendly. Could the authors elaborate on how it can be efficiently implemented in hardware?

**Relation To Broader Scientific Literature:**

The work primarily targets neuromorphic applications, significantly reducing deployment costs by eliminating the need for quantization training.

**Theoretical Claims:**

The theoretical analysis focuses on conversion error, and no apparent errors are present in the conclusions. The mathematical formulations appear consistent with existing ANN-to-SNN conversion frameworks.

---

> ### Author Rebuttal · Authors · 2025-04-01
>
> Thank you for your thorough review. Below, we address some key points of your concerns.
>
> ---
>
> ### **Reference Implementation in Hardware**
>
> Compared to traditional rate coding with IF neurons, our method introduces three additional components:  _1. Membrane potential amplification_, _2. Silent period control_ and _3. Handling input and output of negative spikes_.
>
> _Silent period control is efficiently managed by a state register_. When a neuron starts computing, the register outputs 0. After $P$ clock cycles (where $P$ is the silent period length), it switches to 1 and remains there until a reset. This control signal is __shared across multiple neurons__, resulting in minimal hardware overhead.
>
> _Membrane potential amplification is implemented using a simple shift operation_. Since the shift amount is fixed, this can be achieved only by _introducing a grounding line (logic 0) at the least significant bit (LSB)_ of the membrane potential input to the adder. Specifically, the [n-2:0] bits are hardwired to the adder's input [n-1:1], while the LSB of the input is tied to 0. By performing threshold comparison, the system ensures that the residual value does not exceed $2^{n-1}$, thus preventing overflow. As a result, there is no additional cost for the amplification operations.
>
> _Handling negative spikes is straightforward and requires a two’s complement addition_, which can be efficiently performed by the adder. Specifically, the most significant bit (MSB) of the membrane potential is used to determine its polarity. If the MSB is 1, we take the two’s complement and compare it with the positive threshold in the comparator. This approach effectively compares the absolute value of the membrane potential with the threshold, __maintaining a single comparator__, which incurs minimal hardware overhead. Based on the comparison and the MSB value, we can determine both the presence and the polarity of the spike.
>
> __Thus, the only notable addition is the two’s complement unit__. The silent period control and spike emission modules contribute negligible overhead. We provide an illustration of the reference design at [this URL](https://anonymous.4open.science/r/ICML-2025-Rebuttal-0DE7/README.md). _The amplification operation incurs virtually no overhead and constitutes only a small proportion of the total operations_. For details on the proportion, please refer to our response to Reviewer ujyj.
>
> Based on our analysis and experimental results, __our method is hardware-friendly and has minimal impact on energy consumption__.
>
>
>
> ### Our Main Contributions
>
> First, we would like to clarify that we do not use LIF neurons ($\beta<1$); instead, we introduce the novel TSA neuron ($\beta>1$). The difference in $\beta$ leads to distinct weight patterns and using LIF neurons would cause significant conversion errors due to the residual membrane potential at the end. Additionally, we introduce a negative threshold mechanism to further reduce inference latency. For more details, please refer to our response to Reviewer cfNu.
>
> Second, we highlight the differences between our work and that of Stöckl & Maass [1]. Their approach relies on complex neuron designs and increased computational latency to ensure conversion accuracy and support GeLU activation. In contrast, while our method does not support GeLU activation, it maintains network simplicity and significantly reduces inference latency. Moreover, ReLU activation remains the most widely adopted target for conversion. For additional experiments related to latency, please refer to our response to ujyj.
>
> Lastly, we emphasize that our approach leverages stepwise weighting, which incurs __minimal additional cost__ while delivering substantial benefits. Our method is also __flexible__, _as it adheres to the standard ANN-SNN conversion framework, requiring only the replacement of IF neurons with TSA neurons_. This enables our approach to be effectively applied to architectures such as Transformers and tasks like object detection, significantly __reducing the required number of timesteps__. For further details, we have provided additional experiments on these aspects in our response to pam4 for your reference.
>
> [1] Optimized Spiking Neurons Classify Images with High Accuracy through Temporal Coding with Two Spike
>
> ---
>
> If you have any further questions, we would be happy to address them.

---

### Official Review · Reviewer_ujyj · 2025-03-12

**Overall Recommendation:** 2

**Summary:**

In this work, the authors proposed a new neural coding, which is named canonic signed spike (CCS) coding. For the proposed encoding, they also introduced over-fire-and-correct and threshold optimization methods. The proposed coding method can efficiently transmit various information by transmitting information as binary spikes, but the accumulated membrane potential expresses information over time. The authors theoretically proved the correctness of information transmssion of this coding method. According to the authors' experiments, higher accuracies were achieved in image recognition performed with CNN models.

**Claims And Evidence:**

In order for the proposed method in this paper to be useful, the feasibility of implementation in neuromorphic hardware must be discussed. To implement the proposed method in neuromorphic hardware, synchronization between layers is essential. In addition, the operation of Equation 8 greatly deteriorates the advantage of event-based neuromorphic processors. It has a very big disadvantage that each membrane potential must be increased at every time step even if there are no input spikes. It will not be easy to implement in neuromorphic hardware. Is there a solution for this?

**Essential References Not Discussed:**

The proposed method is similar to the method of the paper below in that it expresses information according to the integrated time difference by utilizing temporal information. In addition, it is similar in that it operates by dividing the integration (silent) and firing phases to transfer temporal information between layers. It is necessary to compare and discuss the methods of the papers below.

Temporal-Coded Spiking Neural Networks with Dynamic Firing Threshold: Learning with Event-Driven Backpropagation, ICCV-23

T2FSNN: deep spiking neural networks with time-to-first-spike coding, DAC-20

**Experimental Designs Or Analyses:**

Additional experiments are required to prove the superiority of the proposed method. It seems to be applicable to other tasks besides image classification. What are the experimental results for tasks such as object detection and segmentation? Also, what if it is applied to transformer models other than CNN? What are the experimental results for neuromorphic datasets?

**Methods And Evaluation Criteria:**

There is no detailed description or analysis of the proposed method. In addition to theoretical proof, it is necessary to experimentally show what influence each factor has. In addition, ablation studies and overhead analysis for the proposed method are required for evaluation.

Does the time step include a silent period? The inference procedure is not clearly explained.

If T and the silent period (P) overlap, a dependency occurs between layers, requiring the total time step of TxL. In this case, can we say that the time step took T instead of TxL?

There is a lack of detailed explanation about OFC.

What would be the performance if there were no negative spikes?

What is the ratio of shift operation and input integration in energy consumption analysis?

“the optimal threshold ~ accuracy in rate coding.” (lines 30~33) - Isn’t this the optimal threshold for the proposed CCS coding? In addition, detailed explanation and analysis of the optimal threshold are required.

**Other Comments Or Suggestions:**

I think it would be helpful to have an overall figure of the proposed approach.

**Other Strengths And Weaknesses:**

Please refer to the above comments.

**Questions For Authors:**

Please refer to the above comments.

**Relation To Broader Scientific Literature:**

Yes, I reviewed it along with the manuscript.

**Theoretical Claims:**

There seems to be no major problem with the theoretical claims.

---

> ### Author Rebuttal · Authors · 2025-04-01
>
> Thank you for your thorough review. Below, we address some key points of your concerns.
>
> ---
>
> ### Energy Overhead of Spike Weighting
>
> To achieve nonlinear encoding, we double the membrane potential at each time step. First, we would like to emphasize that __this method enhances encoded information with almost no additional cost__. In our design, the shift amount is fixed for each step, allowing it to be implemented purely through wiring. Specifically, the [n-2:0] bits are hardwired to the adder's input [n-1:1], while the LSB of the input is tied to 0. This eliminates the need for shift registers, and incurs negligible energy consumption. Furthermore, the amplification is performed independently for each neuron, whereas the majority of operations arises from inter-neuron connections (e.g., convolutional or fully connected layers). Therefore, even if all neurons perform a shift at every time step, the overall cost remains minimal.
>
> We provide a breakdown of the amplification operation's contribution to total _operation count_ in the table below. The experiment was conducted with ResNet-18 on CIFAR-10. It can be observed that the operations for weighting are also minimal in number (accounting for just 4% of AC operations). Overall, their impact can be considered negligible.
>
> Timestep|Amp Ops|ACs|MACs|
> :-|:-|:-|:-
> 8|4.42M|108.5 M|14.72M
> 4|2.23M|76.89 M|7.36M
>
> Additionally, we would like to clarify that our approach is indeed quite __hardware-friendly__. We have provided a detailed reference design in our response to Reviewer MkoZ, which we encourage you to check for further details. We also provide an illustration of the reference design at [this URL](https://anonymous.4open.science/r/ICML-2025-Rebuttal-0DE7/README.md).
>
> ### Methodological Details
>
> #### OFC Method
>
> The goal of OFC is to control the residual membrane potential to reduce conversion loss. This is achieved by lowering the firing threshold (causing Over-Firing) and introducing negative spikes (to Correct the excess firing). This method is also applicable to rate coding, as residual membrane potential impacts conversion loss in that setting as well. We addresses the question of how much to lower the threshold by deriving an optimal threshold mathematically. Notably, we have already conducted ablation experiments on negative spikes in Section 5.3, and experimental validation of the optimal threshold is provided in Section 5.4.
>
> #### Inference Procedure of TSA
>
> We provide the pseudocode for TSA’s forward propagation in Algo. 1, where the input spans all timesteps for clarity. However, our actual implementation __employs a pipelined inference process__: TSA processes spikes _at each timestep_, adjusting its behavior based on its local phase (e.g., silent periods).
>
> Due to the pipelined processing, while the total delay from input to output is $P\times L$, each image only occupies $T+P$ timesteps per layer, where $T$ timesteps are used for primary neural computation. We report $T$ in tables as it represents the actual encoding steps per input. This standard is also applied when comparing TTFS coding and [1], with a similar approach found in [2].
>
> To further illustrate efficiency, we report _actual runtime_ below. Although measured on standard GPUs, these results still reflect the pipeline design in inference. We implemented rate coding as a baseline, setting $P=T$ to simulate [1]. Using four 2080Ti GPUs, we validated 50,000 images on ImageNet with VGG-16. Additionally, we processed 2,000 images _one by one_ and averaged the latency to obtain the _inference latency per image_ (LPI).
>
> Coding Scheme|T|P|Acc.|Runtime|LPI|
> :-|:-|:-|:-|:-|:-|
> rate|256|0|70.50%|11816.04s ($19\times$)|1279ms ($9.2\times$)
> CSS|8|8|75.18%|886.31s ($1.43\times$)|801ms ($5.76\times$)
> CSS|8|1|75.17%|621.15s ($1\times$)|139ms ($1\times$)
>
> [1] Optimized Spiking Neurons Classify Images with High Accuracy through Temporal Coding with Two Spike
> [2] Bridging the Gap between ANNs and SNNs by Calibrating Offset Spikes
>
> #### Related Work
>
> The two works you mentioned focus on implementing TTFS coding. Although both approaches involve accumulating information over time, we would like to emphasize that TTFS accumulates information in _a linear manner_ [3], whereas __our approach follows a nonlinear accumulation process__. This key difference allows us to accumulate information more quickly, thereby reducing the required timesteps.
>
> In our paper, we acknowledge that the silent period concept has been used in TTFS coding and [1]. However, we do not claim a contribution to the silent period itself; rather, we aim to minimize it to reduce inference latency. By incorporating the OFC method, we reduce $P$ to 1, whereas in [3,5], $P=T$.
>
> [3] Temporal-Coded Spiking Neural Networks with Dynamic Firing Threshold: Learning with Event-Driven Backpropagation
>
> ---
>
> If you have any further questions, we would be happy to address them.

---

### Official Review · Reviewer_cfNu · 2025-03-22

**Overall Recommendation:** 4

**Summary:**

The paper proposes an implicitly weighted spiking mechanism for direct ANN-to-SNN conversion. The weight of the spikes, $\beta^{T-t}$, is determined by the temporal location $t \in [1,2, \cdots ,T]$ of the spikes, where an earlier spike gets a higher weight than spikes that arrive later, as $\beta > 1$. Further,  the authors use single-bit signed spikes to reduce the approximation error computed with respect to the ANN activation. The empirical evaluations are performed on CIFAR-10 and ImageNet datasets, where the authors compared the proposed method with existing methods, showing a reduction in temporal latency in direct ANN-to-SNN conversion.

**Claims And Evidence:**

Yes.

**Essential References Not Discussed:**

The authors can include the recent publication [1], which uses signed rate encoding to reduce the variance of noise introduced by the input pixels.

[1] Bhaskar Mukhoty, Hilal AlQuabeh, and Bin Gu, Improving Generalization and Robustness in SNNs Through Signed Rate Encoding and Sparse Encoding Attacks, in The Thirteenth International Conference on Learning Representations (2025).

**Experimental Designs Or Analyses:**

The experimental design is sound.

**Methods And Evaluation Criteria:**

The experimental evaluation could be extended to the CIFAR-100 dataset.

**Other Comments Or Suggestions:**

None.

**Other Strengths And Weaknesses:**

Since the ANN-to-SNN methods pre-suppose the existence of an ANN model, it can be difficult to apply such a method to a neuromorphic dataset where no ANN model exists, or ANN models are equally challenging to train due to the inherent temporal dimension of data.

**Questions For Authors:**

Q1: How do the present neuronal dynamics compare to LIF dynamics?

Q2. Can the experimental evaluations be extended to CIFAR-100?

Q3: What are the challenges to applying the method on neuro-morphic datasets, such as N-NMIST, N-Caltech, and DVS-CIFAR-10?

**Relation To Broader Scientific Literature:**

The paper is well referenced.

**Theoretical Claims:**

The theoretical claims made in the paper are supported with detailed derivations.

---

> ### Author Rebuttal · Authors · 2025-04-01
>
> Thank you for your thorough review. Below, we address some key points of your concerns.
>
> ---
>
> ### Comparison with LIF
>
> The neuron dynamics of TSA and LIF can both given by the following equation:
>
> $$u_{i}^{l}[t]=\beta u_{i}^{l}[t-1]+z_{i}^{l}[t]-S_{i}^{l}[t]$$
>
> Apart from the difference in handling negative spikes, the key distinction between TSA and LIF lies in the choice of $\beta$. While __both mechanisms serve to weight the input__, TSA sets $\beta>1$, resulting in a weight pattern that __decreases__ over time. This design is primarily motivated by two factors:
>
> 1. Enabling _rapid transmission_ of most information.
> 2. Reducing the weight of the final residual information, which is _crucial for conversion accuracy_.
>
> For comparison, we set $\beta = 0.5$ in the table below and observed a significant increase in conversion error. We conducted experiments using VGG-16 on the CIFAR-10 dataset. It can be observed that when using (Ternary) LIF neurons, it is necessary to extend the length of the silent period to reduce conversion loss, which significantly impacts output latency. This further highlights the importance of adopting a decreasing weight pattern.
>
> Neuron|Timestep|Silent Period|Acc.
> :-|:-|:-|:-
> TSA|8|1|96.68%
> LIF|8|1|84.19%
> LIF|8|4|95.32%
> LIF|8|8|96.16%
>
> ### Additional Experiments
>
> Our main contribution is compressing the timesteps for conversion through a stepwise weighting mechanism, which is both convenient and flexible: it still follows the standard ANN-SNN conversion framework, requiring only the replacement of IF neurons with TSA neurons. Therefore, __our method is applicable to a wide range of network architectures, datasets, and tasks__. We have included additional experimental results on _CIFAR-100_ in the table below. The experiments were conducted based on the full-precision VGG-16.
>
> Method|ANN Acc.|Coding Scheme|Timestep|SNN Acc.
> :-|:-|:-|:-|:-
> OPI|76.31%|rate|128|76.25%
> SNN Calibration|77.89%|rate|256|77.68%
> TSC|71.22%|TSC|1024|70.97%
> LC-TTFS|70.28%|TTFS|50|70.15%
> CSS-SNN|76.56%|CSS|8|76.51%
>
>
> Moreover, we have conducted experiments on _object detection tasks_ and applied our encoding method to _Transformer architectures_. The results of these experiments can be found in our response to Reviewer pam4.
>
> Regarding _neuromorphic datasets_, as you pointed out, one of the challenges of applying our method is the absence of an ANN counterpart, which is also a general limitation of ANN-SNN conversion. A possible solution [1] is to integrate temporal information into static features and then train an ANN for classification. We conducted experiments using ResNet18 on the DVS128Gesture dataset, with the results shown in the table below. For comparison, we implemented a simple rate coding as the baseline.
>
> Method|ANN Acc.|Coding Scheme|Timestep|SNN Acc.
> :-|:-|:-|:-|:-
> -|90.94%|rate|128|90.56%
> CSS-SNN|90.94%|CSS|6|90.89%
>
>
> [1] Masked Spiking Transformer
>
> ---
>
> If you have any further questions, we would be happy to address them.

---

### Decision · Program_Chairs · 2025-05-01

**Decision:**

Reject

**Comment:**

The paper proposes a novel Canonic Signed Spike (CSS) coding scheme and an Over-Fire-and-Correct (OFC) method to enhance encoding capacity under reduced timesteps in ANN-to-SNN conversion. The aim is to improve the accuracy of rate-based encoding for spiking neural networks. The authors evaluate their methods on CIFAR-10 and ImageNet, reporting improved performance and demonstrating the potential of their approach in time-constrained scenarios.

After the rebuttal phase, the paper received scores of 4, 2, 2, and 2. Across the reviews, the strengths of the paper include well-organized theoretical derivation, novel methodology, and acknowledged performance improvement. These contributions are particularly pointed out by reviewers MkoZ and pam4. On the other hand, several common concerns were raised: insufficient comparison and discussion of related work, a lack of clarity on the applicability of the method to other datasets (e.g., CIFAR-100, neuromorphic datasets) and architectures, and limited discussion on neuromorphic hardware deployment and its implications.

In the rebuttal, the authors addressed the dataset generalization issue raised by Reviewer cfNu, provided clarifications on related literature requested by Reviewer MkoZ, and offered a pipeline-based implementation strategy in response to Reviewer ujyj' s deployment concerns. However, I think their response to reviewer ujyj did not fully resolve the reviewe'r's comments. Given the overall scores and the fact that all reviewers called for substantial improvements in comparison and analysis, I consider the paper to be on the borderline and recommend rejection. I encourage the authors to revise the paper by enhancing related work discussion, expanding experimental validation, and addressing neuromorphic deployment in more detail.